# Air Drep—A Retrospective Study Evaluating the Influence of Weather Conditions and Viral Epidemics on Vaso-Occlusive Crises in Patients with Sickle Cell Disease Living in French Guiana

**DOI:** 10.3390/ijerph16152724

**Published:** 2019-07-31

**Authors:** Marie-Claire Parriault, Claire Cropet, Aniza Fahrasmane, Stéphanie Rogier, Michaël Parisot, Mathieu Nacher, Narcisse Elenga

**Affiliations:** 1Centre d’Investigation Clinique Antilles Guyane, CIC INSERM 1424, Andrée Rosemon General Hospital, Cayenne 97300, French Guiana; 2Sickle Cell Disease Center, Andrée Rosemon General Hospital, Cayenne 97300, French Guiana; 3Equipe EA3593, Ecosystèmes Amazoniens et Pathologie Tropicale, University of French Guiana, Cayenne 97300, French Guiana; 4Department of Pediatrics, Andrée Rosemon General Hospital, Cayenne 97300, French Guiana

**Keywords:** sickle cell disease, vaso-occlusive crisis (VOC), climate, flu outbreak, French Guiana

## Abstract

(1) Objectives: French Guiana is the French territory most affected by sickle cell disease (SCD). This study investigates the associations between different environmental factors relative to climate, infectious outbreaks, and emergency visits or weekly hospital admissions for vaso-occlusive crisis (VOC). The identification of risk factors would lead to better patient care and patient management, and more targeted prevention and therapeutic education for patients with SCD in French Guiana. (2) Methods: This study was performed using data collected from the medicalized information system and emergency medical records of Cayenne General Hospital, between 1 January 2010 and 31 December 2016. ARIMA models were used to investigate the potential impact of weather conditions and flu epidemics on VOC occurrence. (3) Results: During the study period, 1739 emergency visits were recorded among 384 patients, of which 856 (49.2%) resulted in hospitalization, 811 (46.6%) resulted in hospital discharge, and 72 (4.2%) in another orientation. Decreased temperature and decreased humidity were both independent factors associated with an increase of VOC cases (*p* = 0.0128 and *p* = 0.0004, respectively). When studying severe VOC (leading to hospitalization, with or without prior emergency visit), 2104 hospital admissions were recorded for 326 patients. The only factor associated with severe VOC, in the multivariate analysis, was flu epidemics (*p* = 0.0148). (4) Conclusions: This study shows a link between climate, flu epidemics, and VOC in French Guiana. Patient’s awareness of risks related to climate and flu epidemics should be encouraged, as home prevention measures can help avoid painful crises. Moreover, physicians should encourage patients to get immunized for influenza every year.

## 1. Introduction

Sickle cell disease (SCD) is one of the most common genetic disorders in the world. The main clinical manifestations of this disease, which vary from person to person and change over time, include intermittent and recurrent acute severe pain episodes, as a result of vaso-occlusion. These painful crises frequently require hospital admissions and can lead to serious complications [1,2]. Vaso-occlusive crisis (VOC) is caused by a complex series of processes that are initiated by the polymerization of deoxygenated sickle hemoglobin (HbS) [3]. There is a great variability in the expression of the disease. Some people have no painful crises, while others may have six or more per year [4]. Several factors that may promote VOC have been identified, but there are still many unknown factors. Among these factors, the environment might play a role in the occurrence of VOC [5]. Indeed, several studies have already identified climate and infectious diseases as factors associated with painful crises [3,5,6,7]. French Guiana is the French territory most affected by sickle cell disease. The population, of African ancestry, consists mainly of three groups: Guianese Creoles, Maroons (descendants of runaway slaves), and, more recently, Haitian immigrants [8]. The estimated incidence at birth is one in 227, and the overall frequency of hemoglobin AS (HbAS, made up of normal hemoglobin A and sickle hemoglobin S) carriers is 10% [9]. The major SCD groups include the three main genetic forms that combine different structural hemoglobin variants or thalassemia syndromes (hemoglobin S HbS, hemoglobin C HbC, β-thalassemia): HbSS (68%), HbSC (25%), and Sβ thalassemia (7%) [10]. Identifying a link between environmental factors and VOC hospital admissions would improve patient care and patient management in the hospital, and aid in more targeted prevention and therapeutic education for patients with SCD in French Guiana. This study investigates the associations between different environmental factors, such as: climate, infectious outbreaks, and weekly emergency visits and hospital admissions for VOC.

Ethical Approval: This study was presented at the Cayenne General Hospital Ethical Committee (Number 1-2017-V2) and the database was declared at the Commission Nationale Informatique et Liberté (CNIL, Number 2087477). Patients were informed of the utilization of their data with an informative poster in the medical units concerned with sickle cell disease.

## 2. Materials and Methods

This study was performed using data collected from the medicalized information system program (MISP) and Emergency Medical Records (EMR) of Cayenne General Hospital. Hospital admissions and emergency visits for VOC in SC patients who lived in Cayenne and the surrounding area, between 1 January 2010 and 31 December 2016, were considered for the analysis.

The weather conditions of Cayenne and the surrounding area in the study period were provided by Météo France, the national meteorological and climatological organization. The data included minimum, maximum, and mean measurements for: Daily temperature, daily humidity, and daily cumulative precipitation. The regional health monitoring unit, CIRE Guyane, provided the weekly number of cases of flu.

*Data analysis*—The analysis used weekly data for each time series. First, time series were plotted to look for macroscopic trends. Then, ARIMA models were used to study the potential impact of weather conditions and flu epidemics on VOC occurrence [11]. ARIMA models allow time series modelling, which takes into account potential residual autocorrelations, under the hypothesis that each measure may be correlated to the previous ones. Previous values are also called ‘lagged’ values, with ‘lag *i’* representing the value recorded *i* week(s) ago, when the unit used is the week. Such models allow the combination of three types of temporal processes. The contribution of each temporal process is specified in the notation ARIMA (p,d,q), where p is the order of the autoregressive process AR(p), d the integrated part corresponding to the degree of differencing I(d), and q the order of the moving average model MA(q). ARIMA models including a seasonal trend are specified as ARIMA (p,d,q)(ps,ds,qs).

*Identification stage for the VOC series*—Based on a descriptive analysis (plots) of the weekly number of VOC, and of the series’ autocorrelation function (ACF), and partial autocorrelation functions (PACF), different ARIMA models were tested.

*Estimation and diagnostic checking stage for the VOC series*—Models with significant parameters at the 10% level and where the residual series was Gaussian white noise were retained. The Ljung Box Q test was used to check the non-autocorrelation of residuals of the selected models. Normality was examined graphically based on the histogram of residuals and on the normality plot. Akaike’s Information Criterion (AIC), the principle of parsimony, and the control of the model variance were the performance criteria used to decide for the final model for VOC occurrence, among the potential models retained.

*Testing of explanatory series*—The explanatory series were a weekly average of the minimum, maximum, and mean measurements of daily temperature, and daily humidity. As well as, the weekly average of cumulative precipitation per day and weekly number of flu cases. Lags of 0 (representing the parameter value of the current week) and lag of 1 week (representing the parameter value of the previous week) of each explanatory series were individually integrated in the ARIMA model of VOC occurrence and retained only if significant at the 10% level. In the case of several significant lags for a given series or for different measures of the same series, clinical relevance and control of variance were used to select the parameter leading to the best model.

*Final model*—The explanatory series retained at the end of the previous stage were simultaneously integrated to the VOC model. A backward selection procedure was then used to retain, within the final model, only parameters significant at a 5% level. The Ljung-Box Q test, the residuals histogram, and the normality plot were used to confirm that the residual series of the final model were Gaussian white noise.

## 3. Results

*Description of the different series*—Between 1 January 2010 and 31 December 2016, 1739 emergency visits were recorded among 384 patients, of which 856 resulted in hospitalization, 811 in a hospital discharge, and 72 in another orientation. There were 2104 hospitalizations (with or without prior emergency visit) for VOC recorded over the study period. These hospitalizations involved 326 patients. The average length of the hospital stay was 4.4 days (+/−6.2).

The weekly series of weather parameters (precipitation, temperature, and humidity) highlighted the two seasons of the equatorial, humid climate of French Guiana. The dry season runs from July to December and is marked by low rainfall and low humidity for the region (sometimes approaching 50%) and high temperatures. The rainy season, from mid-December to June, is marked by heavy rainfall, high humidity and slightly lower temperatures. The rainy season is usually interrupted in March for 3 to 4 weeks by a brief return of the dry season. This phenomenon, locally known as the ‘short summer of March’, occurs when the intertropical convergence zone reaches its southern most position, where it parks for a few weeks before returning to the north.

The weekly series of influenza outbreaks clearly showed the occurrence of several infectious episodes during the period of interest. Data showed that except for the year 2008, influenza outbreaks occurred every year. The magnitude of the epidemics was highly variable across years. A major peak was observed in the year 2009. The years 2012 and 2014 also recorded important epidemics. The timing of the influenza outbreaks was also variable across years; although in most cases an intense activity is observed during the first months of the year, it was difficult to identify a clear seasonality.

*ARIMA modeling of VOC*—The weekly series of emergency visits for VOC (Figure 1) was modeled. There was an auto-correlation of residuals, which justified the use of an ARIMA model. Several models were tested. The retained model was an AR(2). The performances of the selected model were correct with parameters that were all significant below 5%. Residuals were uncorrelated and follow a normal distribution.

The weekly series of hospitalizations for VOC (Figure 2) were also modeled. Residuals were auto-correlated, thus an ARIMA model was used. After testing for different models, an AR(4) model was retained. The performances of the selected model were correct with parameters that were all significant below 5%. Residuals were uncorrelated and follow a normal distribution.

*VOC and associated factors*—The correlations between each of the potential explanatory series and the VOC series were examined in bivariate analysis.

Influenza (lag 0 and lag 1), cumulative rainfall (lag 1), maximum humidity (lag 0 and lag 1), minimum temperature (lag 1), maximum temperature (lag 0 and lag 1), and mean temperature (lag 0 and lag 1) were significantly associated with the series of emergency visits for VOC. The resulting bivariate models show uncorrelated residuals with a normal distribution. The explanatory series were thus included in the multivariate model of emergency visits for VOC. The bivariate analysis of the weekly series of hospitalizations for VOC identified flu (lag 1) and mean humidity (lag 0) as associated series, and were thusly selected for the multivariate model.

*Final models*—The final multivariate model of emergency visits for VOC series (Table 1) included the autoregressive parameter (order 2), the mean temperature series (lag 0), and the maximum humidity series (lag 0) as explanatory series. In this model, the negative values of the estimators for the mean temperature series and for the maximum humidity series, indicated:
an increase of the number of painful crises when the average temperature decreased, regardless of the degree of humidityan increase of the number of painful crises when the maximum humidity decreased, regardless of the temperature.

The residuals of the final model were uncorrelated and followed a normal distribution.

The final multivariate model of hospitalizations for VOC (Table 2) included the autoregressive parameters of order 1, 3 and 4 and the influenza series (lag 1). The value of the influenza variable estimator translates one new hospitalization case for VOC every 500 cases of influenza.

## 4. Discussion

Our study shows the temporal variability of the frequency of vaso-occlusive crises among SCD patients, which could be partially explained by environmental factors. Indeed, emergency visits for VOC were influenced by climatic factors, with a decrease in humidity leading to an increase in emergency visits for VOC. This link has already been described in the literature and is explained by a greater risk of dehydration [12,13]. Furthermore, this decrease in humidity occurs mainly in the dry season when the sun is stronger. This is because the skin is exposed and sweating, and therefore, can be more vulnerable to sudden changes in temperature (air conditioner wind, etc.) [12]. A decrease in mean temperature was also associated with an increase in emergency visits for VOC in our study. This association was already described elsewhere and lies in the fact that a decrease in temperature leads to skin cooling, followed by vasoconstriction of the peripherical vessels, and reduction in the velocity of blood circulation [14]. The red blood cells then spend a longer time in a deoxygenated circulation zone, which causes the polymerization of hemoglobin S and the occurrence of a VOC [15]. Although climatic variations are apparently not as marked in tropical environments as in temperate zones, they are sufficient to increase the number of painful crises leading to an emergency visit. In our tropical context where temperatures are mostly hot throughout the year, this may be overlooked by patients and sometimes by their physicians. Intuitively, the rainy season would be the riskier season because it can be cooler when the sky is cloudy and the rainfall is abundant. Indeed, there is a link with this rainy season and VOC. However, counter-intuitively what we show is that the dry season may also be associated with a higher risk of VOC. The present results suggest that this message should be reemphasized for patients and health care professionals, and that patients should remain vigilant in the hotter dry season.

When looking more specifically at severe painful crises that required hospitalization (with or without prior emergency visit), the associated factors were different and influenza was the only explanatory series that correlated significantly with VOC in the multivariate analysis. This association has already been mentioned in the literature, but is rarely tested [16]. The fact that flu causes more serious and frequent complications among SCD patients (severe pain crisis, pneumonia, acute chest syndrome, etc.) explains, in part, the increase in hospitalizations for VOC during influenza epidemics [16]. The question of vaccination also arises here. Protection against infections is a major goal of any care program for patients with SCD. The seasonal flu vaccine is reimbursed by health insurance in France. The vaccine coverage of SCD patients against influenza in France is not exactly known, but it is estimated to be around 30% in French Guiana according to an informal survey of physicians. A study in the UK showed an immunization against viral influenza of 12% in adults and 8% in children [17]. In addition, influenza, by altering the bronchial mucosa, often paves the way for subsequent pneumococcal infections [18]. Patients with sickle cell disease are normally vaccinated, but vaccine valences do no cover all pneumococcal serotypes. Thus, maximizing pneumococcal vaccine coverage should be ensured for all patients.

Our study has several limitations. First of all, the data about VOC were retrospectively collected from medical files that are not intended to be analyzed in a research context, which implies caution should be taken when regarding data quality. For this reason, we started from the year 2010 because the data collection was more complete thereafter. The study period was correct for this type of analysis, but ideally could have been a little longer. The study did not have access to more detailed clinical data in order to attempt to better understand the most likely chain of events leading to VOC. Future studies should aim to improve the description of context, and clinical signs and symptoms in the medical records, in order to conduct more detailed analyses.

Some explanatory series could have been added, in particular pollution measures. It has already been demonstrated that the exposure to air pollution can increase complications and hospitalizations among patients with SCD [3,6,19]. However, these parameters could not be tested in our study because the quality of the data was not good enough to be integrated into the model. The amount of missing data related either to a failure of the measuring device or to defective measures was substantial. The integration of chikungunya and Zika epidemics as explanatory series of VOC would probably have been consistent and relevant. Indeed, graphically, the link between hospitalization for VOC series and these outbreaks is quite clear, but so far there has been only one chikungunya epidemic and one Zika epidemic in French Guiana, which does not allow a time series analysis. The impact of dengue fever should also be tested, but so far the two outbreaks, one in 2010 and the other in 2013, do not provide enough data for analysis. It seemed, however, that graphically there were no correlations between the dengue fever series and the two VOC series.

In conclusion, patients should be aware of increased risks during flu epidemics, low temperature episodes, and low humidity episodes. Therapeutic patient education should enable people with SCD to manage the increased risk of VOC in the presence of aggravating factors. Home prevention measures, such as maintaining good hydration and averting excessive temperature differences, even in the tropical context of French Guiana, could avoid some painful crises. Moreover, physicians should encourage patients to get immunized for influenza vaccine every year. Improving the influenza vaccine coverage and optimizing pneumococcal vaccine coverage could reduce the number of hospitalizations and admissions for VOC during the flu outbreaks. It is essential that patients know their disease, as well as the mechanisms that underlie it, in order to better adapt their behaviors. This is the central principle of therapeutic patient education. It therefore seems important to communicate these results to patients, but also to their entourage and to physicians, to improve knowledge about the sickle cell disease and its management.

## Figures and Tables

**Figure 1 ijerph-16-02724-f001:**
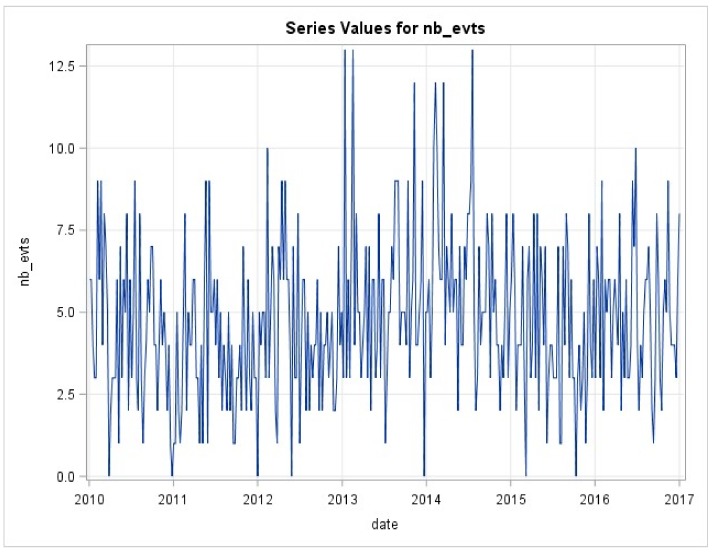
Weekly series of emergency visits for vaso-occlusive crisis (VOC).

**Figure 2 ijerph-16-02724-f002:**
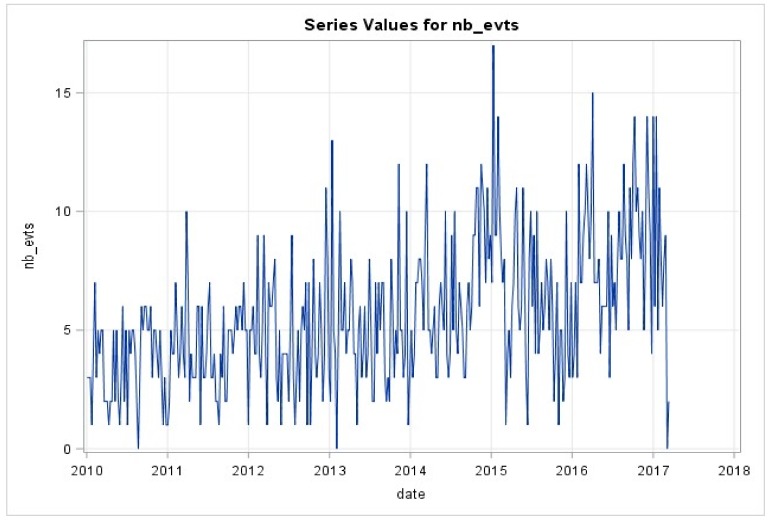
Weekly series of hospitalizations for VOC.

**Table 1 ijerph-16-02724-t001:** Parameters estimates of final multivariate model of emergency visits for VOC.

Parameters	Estimate	Standard Error	*p* Value
Constant	36.959	6.999	<0.0001
VOC (lag 2)	0.102	0.052	0.054
Mean temperatures (lag 0)	−0.478	0.191	0.0128
Maximum humidity	−0.201	0.056	0.0004
lag 0 represents the value zero week before the VOC			
lag 2 represents the value two weeks before the VOC			

**Table 2 ijerph-16-02724-t002:** Parameter estimates of final multivariate model of hospitalizations for VOC.

Parameters	Estimate	Standard Error	*p* Value
Constant	4.719	0.508	<0.0001
VOC (lag 1)	0.24	0.05	<0.0001
VOC (lag 3)	0.223	0.051	<0.0001
VOC (lag 4)	0.227	0.053	<0.0001
Influenza cases (lag 0)	0.0018	0.0003	0.0148
lag 0 represents the value zero week before the VOC	
lag 1 represents the value one week before the VOC	
lag 3 represents the value three weeks before the VOC	
lag 4 represents the value four weeks before the VOC

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
