# Peer review of "Air Drep—A Retrospective Study Evaluating the Influence of Weather Conditions and Viral Epidemics on Vaso-Occlusive Crises in Patients with Sickle Cell Disease Living in French Guiana"

_ijerph, 2019, doi:10.3390/ijerph16152724_

Round 1
Reviewer 1 Report
This study presents a retrospective study which investigates various environmental and public health factors and their relationship to vaso-occlusive crises in sickle cell disease. The analysis presented in the paper seems to indicate that frequency of VOC may be related to climatic factors such as humidity, high temperatures, and exposure to sunlight. The authors suggest that these factors may lead to patients becoming dehydrated leading to VOC. While this appears to be a logical conclusion, the authors of the study offer no clinical data, found in the records they had access to, which might indicate that patients are suffering from dehydration. Without further patient data, it is extremely difficult to put their well-done statistical analysis into any sort of context.
The second part of their analysis presents a statistical link between influenza infection and the onset of VOC and resultant hospitalization. This general observation in and of itself is not novel, given the ample literature supporting this claim, however the limited retrospective study does add to evidence to these generic observations.
Reviewer 2 Report
This is an interesting study that examined the effect of weather conditions as well viral infections on VOE in patients with SCD. Overall, the manuscript is well-written with enough details in different sections. I have few comments for the authors to consider:
- The authors should define in their figures what lag1, lag3 and lag4 means
- The authors can elaborate more on directions for future research and how these data can help to educate patients to better prevent or avoid VOE under certain weather circumstance
- The authors can expand more on study limitations given some inherent in the study design and data source
